# Two New Compounds from the Fungus *Xylaria nigripes*

**DOI:** 10.3390/molecules28020508

**Published:** 2023-01-04

**Authors:** Hongping Long, Siqian Zhou, Lanqing Li, Jing Li, Jikai Liu

**Affiliations:** 1The First Hospital of Hunan University of Chinese Medicine, Center for Medical Research and Innovation, Changsha 410007, China; 2School of Pharmaceutical Sciences, South-Central Minzu University, Wuhan 430074, China; 3Department of Pharmacy, Xiangya Hospital, Central South University, Changsha 410008, China; 4National Clinical Research Center for Geriatric Disorders, Xiangya Hospital, Central South University, Changsha 410008, China

**Keywords:** *Xylaria nigripes*, structural elucidation, amino acid, neuroprotective activities

## Abstract

In the process of discovering more neural-system-related bioactive compounds from *Xylaria nigripes*, xylariamino acid A (**1**), a new amino acid derivative, and a new isovaleric acid phenethyl ester (**2**) were isolated and identified. Their structures and absolute configurations were determined by analyses of IR, HRESIMS, NMR spectroscopic data, and gauge-independent atomic orbital (GIAO) NMR calculation, as well as electronic circular dichroism (ECD) calculation. The isolated compounds were evaluated for their neuroprotective effects against damage to PC12 cells by oxygen and glucose deprivation (OGD). Compounds **1** and **2** can increase the viability of OGD-induced PC12 cells at all tested concentrations. Moreover, compound **2** (1 μmol L^−1^) can significantly reduce the percentage of apoptotic cells.

## 1. Introduction

Medicinal fungi are important sources of physiologically beneficial molecules, and many of them are considered to be functional foods. Various types of secondary metabolites with interesting biological activities have been isolated from medicinal fungus, which have the potential to be used as valuable chemical resources for drug discovery [1,2,3]. *Xylaria nigripes* (family Xylariaceae) is a medicinal and edible fungus with fruiting bodies well known as “Wu Ling Shen” or “Lei Zhen Zi” in China [4,5,6]. As a traditional Chinese medicinal fungus, *X. nigripes* has been used to treat insomnia, trauma, depression, mental illness, and diseases of the nervous, immune, and endocrine systems [5,6,7,8,9,10,11,12].

Previous investigations on the metabolites of *X. nigripes* resulted in the isolation of secondary metabolites including alkaloids, sesquiterpenes, steroids, and polyketides, and some of them exhibited promising anti-inflammatory, antioxidant, neuroprotective, and cholesterol ester transfer protein inhibition activities [13,14,15,16,17,18,19]. In our previous research, naphthalenone derivatives and resorcinol derivatives were isolated from *X. nigripes*, which had potential in protecting PC12 cells from OGD-induced injury in vitro [20,21]. In order to search for secondary metabolites with neuroprotective activity from medicinal fungus, the extract of the rice fermentation of *X. nigripes* was further investigated. As a result, a new amino acid derivative, named xylariamino acid A (**1**), and an isovaleric acid phenethyl ester (**2**) were isolated (Figure 1). The structures and absolute configurations of the new compounds were established unambiguously by spectroscopic data analysis, as well as ^13^C NMR chemical shifts and electronic circular dichroism (ECD) calculations. Herein, the isolation, structure elucidation, and biological activities of compounds **1** and **2** are described.

## 2. Results and Discussion

Xylariamino acid A (**1**) was isolated as a colorless powder. Its molecular formula was determined to be C_17_H_25_NO_4_ by the positive HRESIMS (*m/z* 330.16748 [M + Na]^+^, calcd for C_17_H_25_NO_4_Na, 330.16758), indicating six degrees of unsaturation. The 1H NMR spectrum (Table 1) revealed five aromatic protons at *δ*_H_ 7.31 (2H, m, H-2/6), 7.26 (1H, m, H-4), and 7.24 (2H, m, H-3/5), owing to a monosubstituted benzene ring; one oxygenated methylene proton at *δ*_H_ 4.08 (2H, dd, *J* = 7.2, 6.6 Hz); two methylenes at *δ*_H_ 3.12 (2H, m), 1.46 (1H, m), and 1.58 (1H, m); three methine protons at *δ*_H_ 3.80 (1H, t, *J* = 6.6 Hz), 3.33 (1H, m), and 1.70 (1H, m); and three methyls at *δ*_H_ 1.12 (3H, t, *J* = 7.2 Hz), 0.89 (3H, d, *J* = 6.6 Hz), and 0.86 (3H, d, *J* = 6.6 Hz). The ^13^C NMR spectrum displayed 17 carbon resonance signals, which were assigned with the aid of DEPT and HSQC spectroscopy as six aromatic carbons (*δ*_C_ 130.4 × 2, 129.7 × 2, 128.2, 137.4), three methyls (*δ*_C_ 23.0, 22.8, 14.2), three methylenes including one oxygenated resonance (*δ*_C_ 62.7, 42.5, 38.8), three methines (*δ*_C_ 62.6, 60.9, 25.9), and two ester carbonyls (*δ*_C_ 176.5, 173.1).

Further analyses of the 1H–1H COSY and HMBC spectra led to the elucidation of the planar structure of compound **1** (Figure 2). The 1H–1H COSY spectrum indicated the presence of four spin–spin coupling systems: H-2/H-3/H-4/H-5/H-6, H-7/H-8, H-10/H-11, and H-12/H-13/H-14/H-15/H-16. The HMBC correlations of H-11 to C-10, H-10 to C-9, H-8 to C-1, H-7 to C-9, and H-2 to C-7 revealed the presence of a substituted ethyl 3-phenylpropionate unit. The remaining NMR resonances were attributed to an isocaproic acid unit according to the simultaneous HMBC correlations of H_3_-15/H_3_-16 to C-13 and C-14, and the key HMBC correlations from H-14 to C-12, as well as from H-13 to C-17. Considering the molecular formula and the downfield chemical shifts of C-12 (*δ*_C_ 60.9) and C-8 (*δ*_C_ 62.6), the two units were deduced to be connected with a nitrogen atom between C-12 and C-8, which was supported by the HMBC correlation from H-12 to C-8. Thus, compound **1** was assigned as an amino acid derivative as shown in Figure 1.

In order to further confirm the relative configuration of compound **1**, ^13^C NMR chemical shift calculations were performed on compound **1** to specify the configurations of C-8 and C-12. All four reasonable configurations of compound **1** were labeled as 8*R*,12*S*—**1a**; 8*R*,12*R*—**1b**; 8*S*,12*S*—**1c**; and 8*S*,12*R*—**1d** (Figure 3), and their chemical shift calculations were performed using the gauge-independent atomic orbital (GIAO) method at the B3LYP-D3(BJ)/TZVP (IEFPCM, CH_3_OH) level of theory, following a reported STS protocol [20]. The results showed that the calculated data of **1b**/**1c** have the lowest mean absolute error (MAE) and root mean square (RMS) values, and the *P*_mean_ and *P*_rel_ parameters showed that **1b** or **1c** is the more reasonable configuration (Table 2). The relative configuration of compound **1** was thus established as 8*R**,12*R**. Moreover, the absolute configuration of compound **1** was determined by a comparison of the experimental ECD and calculated ECD values. The calculated ECD spectrum of 8*R*,12*R*—**1** showed negative Cotton effects in the wavelength ranging from 200 to 240 nm, which matched well with the experimental curve (Figure 4), allowing the absolute configuration of compound **1** to be determined as 8*R*,12*R*. Consequently, the structure of compound **1** was unequivocally established and named xylariamino acid A.

Compound **2** was obtained as yellow oil. Its molecular formula of C_13_H_18_O_4_ was established by the positive HRESIMS ion at *m/z* 261.1098 [M + Na]^+^ (calcd for C_13_H_18_O_4_Na, 261.1097), corresponding to five degrees of unsaturation. The 1H NMR spectrum of compound **2** revealed one set of AA’BB’ coupling benzene protons at *δ*_H_ 7.05 (2H, d, *J* = 8.5 Hz, H-2/6) and 6.72 (2H, d, *J* = 8.5 Hz, H-3/5); one oxygenated methine proton at *δ*_H_ 3.90 (1H, d, *J* = 4.8 Hz); a methine proton at *δ*_H_ 1.96 (1H, m); one oxygenated methylene at *δ*_H_ 4.29 (2H, m); one methylene at *δ*_H_ 2.85 (2H, t, *J* = 6.6 Hz); and two methyl groups at *δ*_H_ 0.92 (3H, d, *J* = 7.2 Hz) and 0.82 (3H, d, *J* = 7.2 Hz). The ^13^C NMR and HSQC spectra displayed 13 carbon resonances, which could be assigned as follows: one ester carbonyl (*δ*_C_ 175.6), one benzene ring (*δ*_C_ 130.9 × 2, 116.2 × 2, 157.1, 129.8), two methines (*δ*_C_ 33.3, 76.6), two methylenes (*δ*_C_ 35.2, 66.8), and two methyls (*δ*_C_ 19.1, 17.0) (Table 1). The benzene ring and ester groups contribute all the five degrees of unsaturation, indicating that compound **2** is a benzene derivative with linear substitutions.

In the HMBC spectrum (Figure 2), the correlations from H-7 to C-2/6, and from H-8 to C-1, together with the 1H–1H COSY cross-peaks of H-7/H-8, established the structure of a 4-hydroxyphenyl ethanol moiety. In addition, the HMBC correlations from H-13 to C-10/C-11/ C-12, and from H-11 to C-9/C-10, along with the analysis of the 1H–1H COSY spectrum indicated the presence of a 2-hydroxyisovaleric acid moiety. Subsequently, the HMBC correlations from H-8 to C-9 verified that the two moieties were connected through an ester bond between the C-8 and C-9. Thus, the planar structure of compound **2** was established as shown in Figure 1. The absolute configuration of compound **2** was established by a comparison of the experimental ECD and calculated ECD data. The results showed that the calculated ECD spectrum of 10*R*—**2** matched well with the negative Cotton effect at the range from 200 to 240 nm in the experimental spectrum (Figure 5). Thus, the structure with the absolute configuration of compound **2** was unequivocally established as 10*R*.

Compounds **1** and **2** were evaluated for their neuroprotective effects against damage to PC12 cells by OGD. As a result, compounds **1** and **2** increased the viability of OGD-induced PC12 cells at all tested concentrations (Figure 6) and did not exhibit cytotoxicity (Appendix A). As shown in Figure 7, compared with the control group, PC12 cells were treated with hypoxia for 6 h and incubated with or without the test compounds for 24 h, and showed obvious apoptotic morphology. In addition, compared with the model group, pretreatment with nimodipine could reduce early and late apoptotic cells after OGD. Furthermore, compound **2** (1 μmol L^−1^) could reduce the percentage of apoptotic cells compared with those treated with OGD (Figure 7). Due to the scarcity of the compounds, the neuroprotective activity of compounds **1** and **2** has not been further studied, and more assays (ELISA-based, colorimetric, or cytometry) are need to verify their neuroprotective activity.

## 3. Materials and Methods

### 3.1. General Experimental Procedures

UV spectra were afforded with a UH5300 UV-VIS Double Beam Spectrophotometer. ECD spectra were obtained on a Chirascan CD spectrometer (Applied Photophysics, London, UK). HRESIMS spectra were recorded on a Q Exactive Obitrap mass spectrometer (ThermoFisher Scientific, Waltham, MA, USA) and UPLC-ESI-Q-TOF-MS (1290 UPLC-6540, Agilent Technologies Inc., Palo Alto, CA, USA) HRMS spectrometer. IR spectra were obtained with a Shimadzu Fourier Transform Infrared Spectrometer using KBr pellets. NMR spectra were recorded on a Bruker Avance III 600 MHz spectrometer with TMS as an internal standard. Chemical shifts (*δ*) were expressed in ppm with reference to the solvent signals. Column chromatography (CC) was performed on silica gel (200–300 mesh, Qingdao Marine Chemical Ltd., Qingdao, China) and Sephadex LH-20 (Pharmacia Fine Chemical Co., Ltd., Uppsala, Sweden). Medium Pressure Liquid Chromatography (MPLC) was performed on a Biotage SP1 System and columns packed with RP-18 gel. Preparative high-performance liquid chromatography (prep-HPLC) was performed on an Agilent 1260 liquid chromatography system equipped with Zorbax SB-C18 columns (Agilent, 5 μm, 9.4 mm × 150 mm) and a DAD detector. Fractions were monitored by TLC (GF 254, Qingdao Haiyang Chemical Co., Ltd., Qingdao, China).

### 3.2. Fungal Material

The fungus *Xylaria nigripes* (Kl.) Sacc. (Xylariaceae) was collected from Ailao Moutain, Yunnan Province of China in 2013 and was identified by Prof. Yu-Cheng Dai (Beijing Forestry University), a mushroom specialist. The strain of *X. nigripes* in this study was isolated from the fresh fruiting bodies and kept on a potato, dextrose, and agar (PDA) culture medium. A voucher specimen (No. CGBWSHF00611) was deposited at South-Central Minzu University, China.

### 3.3. Fermentation and Isolation

This strain of *X. nigripes* was cultured on potato dextrose agar medium at 25 °C. After eight days, the agar plugs were cut into small pieces to incubate on a solid rice medium in 500 mL Erlenmeyer flasks (100 g rice and 100 mL tap water for each Erlenmeyer flask, the total weight of rice was 7 kg) to culture for a further 30 days at 25 °C. Then, the culture medium and the mycothallus were extracted with ethyl acetate 3 times. The extract was concentrated under vacuum with a rotary evaporator to yield 38 g crude extract.

The crude extract was separated by a silica gel column using petroleum ether and ethyl acetate with a ratio from 1:0 to 0:1 to obtain ten fractions (A–J). Fraction D (8.5 g) was separated by MPLC with a stepwise gradient of MeOH–H_2_O (0–100%) to afford five fractions (D1–D5). Fr. D2 (0.5 g) was purified by prep-HPLC (MeCN/H_2_O, *v*/*v*, from 25/75 to 35/65 in 20 min, flow speed: 4 mL/min) to give compound **1** (5.8 mg, *t*_R_ = 16.5 min). Fraction I (4.0 g) was separated by MPLC with a stepwise gradient of MeOH–H_2_O (0–100%) to afford five fractions (I1–I5). Fr. I2 (0. 8 g) was further separated by Sephadex LH-20 (MeOH) to give eight fractions I2a–I2h. Subfraction I2e (20 mg) was purified by prep-HPLC (MeCN/H_2_O, *v*/*v*, from 20/80 to 30/70 in 20 min, flow speed: 4 mL/min) to give compound **2** (3.9 mg, *t*_R_ = 14.6 min).

Xylariamino acid A (**1**): colorless powder; UV (MeOH) λ_max_(log ε) 210 (1.94) nm; IR (KBr)ν_max_: 3578, 3379, 1736, 1603, 1379, 1356 cm^−1^; 1H NMR (600 MHz, methanol-d_4_) and ^13^C NMR (150 MHz, methanol-d_4_) data, see Table 1; HRESIMS: *m/z* 330.16748 [M + Na]^+^ (calcd for 330.16758).

Compound (**2**): yellow oil; UV (MeOH) λ_max_(log ε) 225 (2.87) nm; IR (KBr)ν_max_: 3368, 2947, 2835, 1651, 1454, 1418, 1113, 1032 cm^−1^; 1H NMR (600 MHz, methanol-d_4_) and ^13^C NMR (150 MHz, methanol-d_4_) data, see Table 1; HRESIMS: *m/z* 261.11012 [M + Na]^+^ (calcd for 261.10973).

### 3.4. Cell Viability Assay

A nRPMI-1640 culture medium and phosphate buffer saline (PBS) were purchased from Gibco Co. (Grand Island, NY, USA). Dimethyl sulfoxide (DMSO) and a cell counting kit-8 (CCK-8) were purchased from Boster Biol. Tech (Wuhan, China). An Annexin V-FITC/PI assay kit was purchased from Beyotime Biotechnology, Ltd. (Shanghai, China). Fetal bovine serum (FBS) and PC12 cells were purchased from Procell Life Science & Technology Co., Ltd. (Wuhan, China). Nimodipine was purchased from Yabao Pharmaceutical Group Co., Ltd. (Shanxi, China).

PC12 cells were seeded into 96-well plates at a density of 1 × 10^4^ cells well-1 and incubated overnight at 37 °C for 24 h. Compounds **1** and **2** were prepared in serial dilutions from 0.01 to 10 µM in a serum-free medium and the cells were treated for 24 h. Then, the medium was replaced with a 10 μL/well CCK8 solution (Bioss, Beijing, China) and the cells were incubated at 37 °C for 1 h. The optical density (OD) was measured at 450 nm by a microplate reader (Enspire, PerkinElmer, Waltham, MA, USA). The neuroprotective activity of the tested compounds was assayed in 96-well plates, according to the methods in the previously published article [18].

### 3.5. Cell Apoptosis Assay

The apoptosis of PC12 cells was detected using the Annexin V-FITC/PI (Beyotime Biotechnology, Shanghai, China). The logarithmically growing human neuroblastoma PC-12 cells were plated at a density of 8 × 10^4^ cells well-1 in 24-well plates for 24 h. Then, cells were treated with hypoxia for 6 h, and then incubated with or without the test compounds for 24 h. A single adherent culture of PC-12 cells was washed with PBS, then the cells were evenly mixed with 195 µL 1× binding buffer, 10 µL Annexin V-FITC, and 10 µL PI and incubated for 15 min in the dark. The apoptosis data were analyzed by ZEN software (Zesis, Jena, Germany).

### 3.6. Computation Methods

The software Crestwas was used to search the conformers of compound **1** on the GFNFF level of theory [21,22], followed by optimization on the GFN2-XTB level with a 4 kcal/mol energy window to remove high energy conformers [23]. The optimization and frequency calculation of each conformer was performed on the B3LYP-D3(BJ)/TZVP (IEFPCM, CH_3_OH) level of theory. The DFT GIAO ^13^C NMR calculation was calculated on the ωB97xD/6-31G* (IEFPCM, CHCl_3_) level, and the data processing followed the reported STS protocol [20]. The calculated shielding tensors of the conformers were Boltzmann-averaged based on Gibbs free energy. The theoretical ECD was calculated by time-dependent density functional theory (TDDFT) at the mPW1PW91/6-311g(d) level with an IEF-PCM solvent model (MeOH) as well. SpecDis v1.71 was used to simulate the ECD curve with a sigma/gamma value of 0.35 eV. The calculated ECD curve of each conformer was Boltzmann-averaged based on their Gibbs free energy. All DFT calculations were performed by the Gaussian 16 software package.

## 4. Conclusions

In this work, two new compounds were isolated from the cultures of *X. nigripes*. The structures and absolute configurations of the isolates were unambiguously identified by a detailed interpretation of the NMR spectroscopic and mass spectrometric data, ^13^C NMR chemical shift, and ECD calculations. The neuroprotective activity test of compounds **1** and **2** showed that both of them can increase the viability of OGD-induced PC12 cells at all tested concentrations. Moreover, compound **2** could reduce the percentage of apoptotic cells at the concentration of 1 μmol L^−1^, which suggested that it may have potential neuroprotective activity.

## Figures and Tables

**Figure 1 molecules-28-00508-f001:**
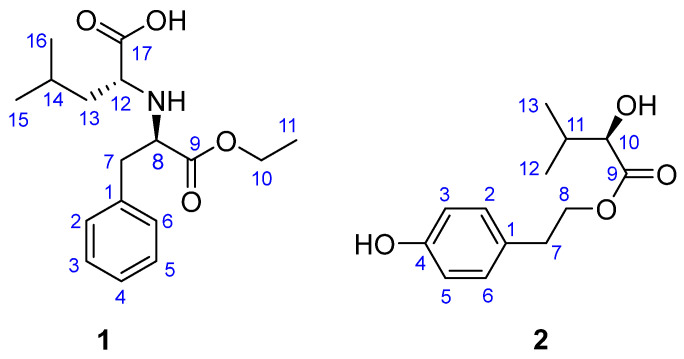
Structures of compounds **1** and **2**.

**Figure 2 molecules-28-00508-f002:**
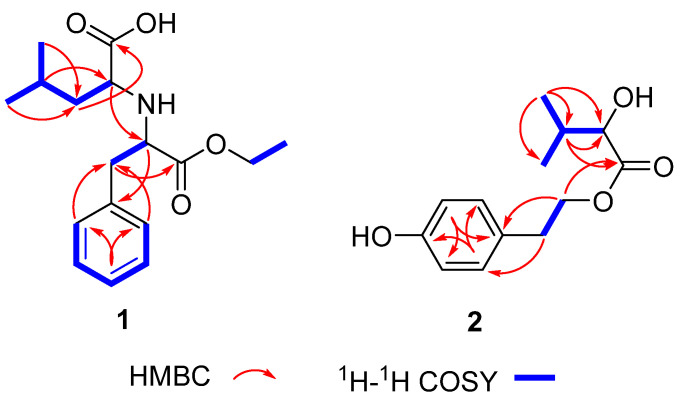
Key 2D NMR correlations of compounds **1** and **2**.

**Figure 3 molecules-28-00508-f003:**
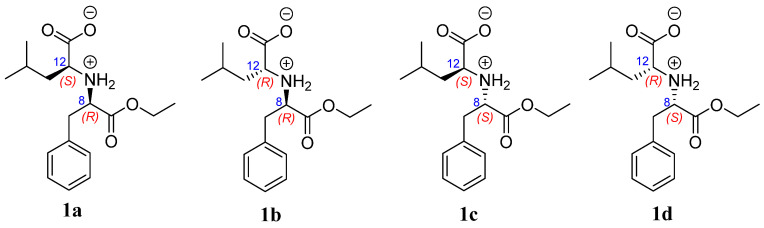
Four possible relative configurations of compound **1**.

**Figure 4 molecules-28-00508-f004:**
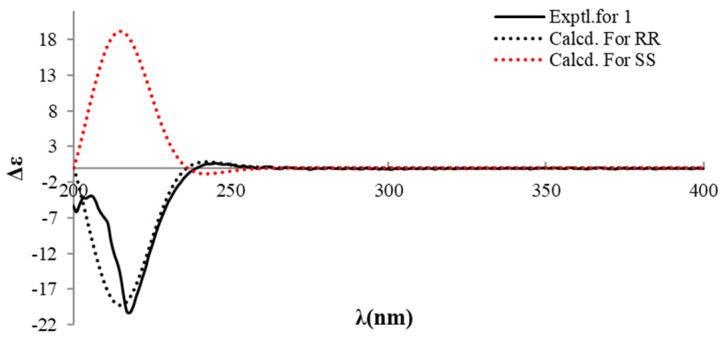
Experimental and calculated ECD spectra of compound **1**.

**Figure 5 molecules-28-00508-f005:**
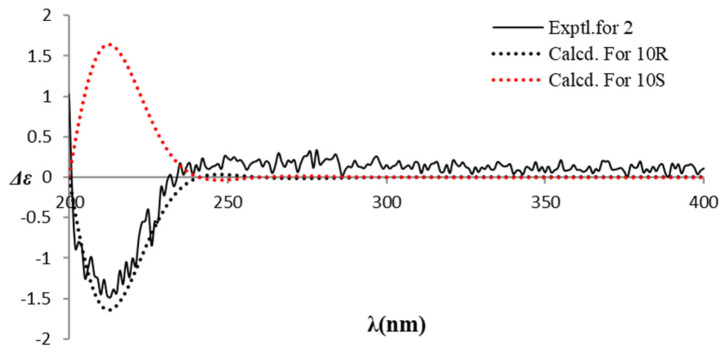
Experimental and calculated ECD spectra of compound **2**.

**Figure 6 molecules-28-00508-f006:**
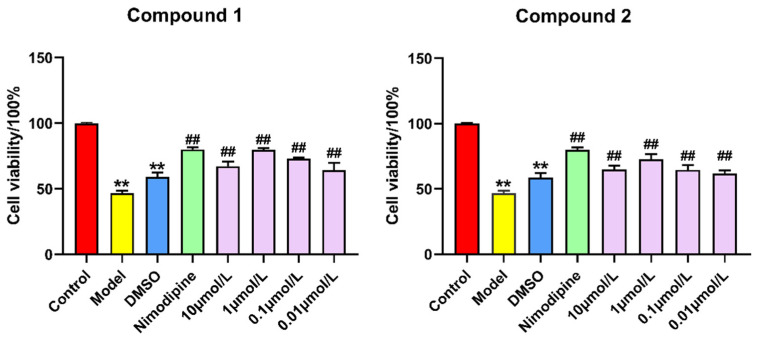
Effects of compounds **1** and **2** on cell viability in OGD-induced PC12 cells. The values represent mean ± SD (*n* = 6). ** *p* < 0.01 vs. the control group; ^##^
*p* < 0.01 vs. the model group.

**Figure 7 molecules-28-00508-f007:**
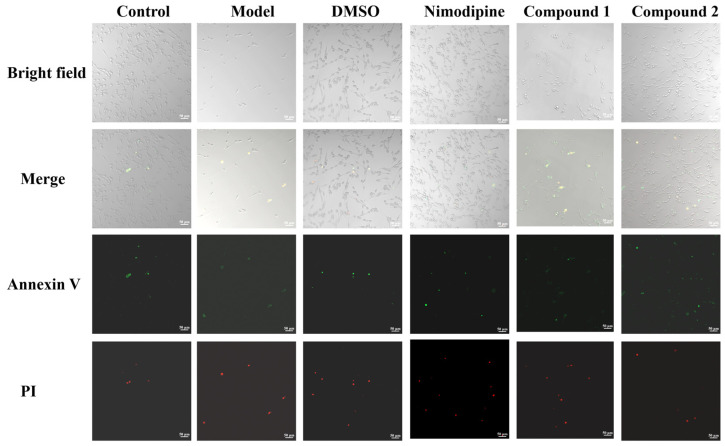
Annexin V-FITC/PI staining and fluorescence microscopy images (original magnification, 200×) (PC12 cells treated with 1 μmol L^−1^ of compounds **1** and **2**).

**Table 1 molecules-28-00508-t001:** 1H NMR (600 MHz) and ^13^C NMR (150 MHz) data of compounds **1** and **2** in methanol-d_4_.

No.	1		2	
	*δ* _H_	*δ* _C_	*δ* _H_	*δ* _C_
1		137.4		129.8
2	7.31, m	129.7	7.05, d (8.4)	130.9
3	7.24, m	130.4	6.72, d (8.4)	116.2
4	7.26, m	128.2		157.1
5	7.24, m	130.4	6.72, d (8.4)	116.2
6	7.31, m	129.7	7.05, d (8.4)	130.9
7	3.12, m	38.8	2.85, t (6.6)	35.2
8	3.80, t (6.6)	62.6	4.29, m	66.8
9		173.1		175.6
10	4.08, dd (7.2,6.6)	62.7	3.90, d (4.8)	76.6
11	1.12, t (7.2)	14.2	1.96, m	33.3
12	3.33, m	60.9	0.92, d (7.2)	19.1
13	1.46, m, 1.58, m	42.5	0.82, d (7.2)	17.0
14	1.70, m	25.9		
15	0.89, d (6.6)	22.8		
16	0.86, d (6.6)	23.0		
17		176.5		

**Table 2 molecules-28-00508-t002:** Calculated ^13^C chemical shifts of **1a**–**1d** fitting with the experimental data of compound **1** following STS protocol.

NO.	Exptl. *Δ*_C_	Calcd. *Δ*_C_ 1a/1d	Dev.	Calcd*. Δ*_C_ 1b/1c	Dev.
1	128.2	127.52	0.68	127.36	0.84
2	130.4	129.05	1.35	128.91	1.49
3	129.7	128.98	0.72	129.30	0.40
4	137.4	137.72	0.32	137.05	0.35
5	129.7	128.98	0.72	129.30	0.40
6	130.4	129.05	1.35	128.91	1.49
7	38.8	40.17	1.37	42.23	3.43
8	62.6	66.37	3.77	64.28	1.68
9	173.1	175.48	2.38	176.41	3.31
10	62.7	63.74	1.04	62.73	0.03
11	14.2	14.85	0.65	14.97	0.77
12	60.9	61.59	0.69	60.30	0.60
13	42.5	41.01	1.49	40.88	1.62
14	25.9	23.97	1.93	25.62	0.28
15	22.8	22.28	0.52	22.21	0.59
16	23	22.28	0.72	22.21	0.79
17	176.5	175.77	0.73	176.12	0.38
		MAE	1.20	MAE	1.09
		RMS	1.46	RMS	1.45
		*P* _mean_	27.01%	*P* _mean_	31.30%
		*P* _rel_	7.56%	*P* _rel_	92.44%

## Data Availability

All data are available in this publication and in the Appendix A.

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
