# Peer review of "Two New Compounds from the Fungus Xylaria nigripes"

_molecules, 2023, doi:10.3390/molecules28020508_

Round 1

Reviewer 1 Report (Previous Reviewer 1)

This manuscript describes the isolation and characterization of new compounds from Xylaria nigripes and their neuroprotective activity.

The chemistry part looks interesting from the natural product point of view but there is insufficient data to support the claimed neuroprotective activities of the isolated compounds.

I am not able to infer from Fig 7 that the compounds reduced the level of apoptosis (due to to the insufficient number of cells in each panel). More data is needed to substantiate that postulation.

Author Response

Response: Thanks for the reviewer’s carefully review. We were actually aware of this problem, especially for the staining, and we had performed the bioassay several times. However, we almost used all pure samples of compounds 1 and 2, and the results were still similar to that presented herein. For the conclusion drawn from Fig 7, we have used less certain expression to avoid possible misunderstanding.

Reviewer 2 Report (New Reviewer)

Congratulation for the good work. It takes new interesting information.

It's just some suggestion:

Abstract: explain better the aim of the paper.

Include Figure 1 before line 42.

Line 44 - Is it just Results or Results and discussion?

Author Response

  1. Abstract: explain better the aim of the paper.

     Response: Modified as recommended.

  1. Include Figure 1 before line 42.

     Response:  Modified as recommended.

  1. Line 44 - Is it just Results or Results and discussion?

     Response:  Modified as recommended.

Reviewer 3 Report (New Reviewer)

1.      The word “new” should be excluded from text (line 37), since isovaleric acid phenethyl ester is a well-known compound.

2.      The difference between the calculated spectrum and the real one should be explained.

3.      The new compound (2) should be characterized in more detail.

Author Response

  1. The word “new” should be excluded from text (line 37), since isovaleric acid phenethyl ester is a well-known compound.

     Response: Modified as recommended.

  1. The difference between the calculated spectrum and the real one should be explained.

     Response: Modified as recommended.

  1. The new compound (2) should be characterized in more detail.

    Response: More details have been added as recommended.

Round 2

Reviewer 1 Report (Previous Reviewer 1)

It seems that the problem lies on the choice of assay. Assays yielding quantitative results could have been chosen (ELISA-based, colorimetric, flow cytometry) to give more convincing results with regard to the potential neuroprotective activity of the compounds.

I agree with the authors that editing is needed, and would like to suggest the authors to consider the following:

1. Provide basis to investigate the bioactivity of these isolated compounds (based on the class of compounds, structure-activity relationship, etc.)

2. If the authors have the data for the crude extracts, that can be used to justify the above.

2. Add more discussion on the potential neuroprotective activity of the compounds and/or compounds from xylaria and/or compounds of similar class

Author Response

  1. Provide basis to investigate the bioactivity of these isolated compounds (based on the class of compounds, structure-activity relationship, etc.)

Response: The major traditional usage of X. nigripes are closely related to nervous system. In our previous research, we found that the naphthalenone derivatives isolated from the crude extracts of X. nigripes have potential neuroprotective effects by enhancing cell viability, affecting the level of oxidative stress and inhibiting apoptosis against OGD-induced injury in PC12 cells. Moreover, we also found that resorcinol derivatives which isolated from X. nigripes have the potential to be developed as neuroprotective agents. We have added more information regarding the basis as recommended.

  1. If the authors have the data for the crude extracts, that can be used to justify the above.

Response: Thanks for the reviewer’s suggests, and I agree on your point of view. However, we don't have the data for the crude extracts at the moment.

  1. Add more discussion on the potential neuroprotective activity of the compounds and/or compounds from xylaria and/or compounds of similar class

Response: Thanks for the reviewer’s carefully review. The discussion on the potential neuroprotective activity of the compounds from xylaria have been addended in the introduction section of the revised manuscript.

This manuscript is a resubmission of an earlier submission. The following is a list of the peer review reports and author responses from that submission.

Round 1

Reviewer 1 Report

In this manuscript, the authors reported on the isolation and characterisation of compounds from Xylaria nigripes and assessment of their neuroprotective activities.

1. The fungal origin of the compounds needs to be demonstrated. The extraction was performed on media containing fungal mycelium, so it will be difficult to confirm the compound indeed originated from fungal biomass and not from the media. The compound could be from the media. This issue was not addressed.

2. The effect of the compounds on cell viability should be demonstrated before the neuroprotective assay was carried out.

3. In the apoptosis assay, the number of cells in different treatments seems to show a large variation. Too little cells showing signals for either PI or AV. How can one then conclude that treatment with compound 2 reduced apoptosis based on Figure 7 (lines 133-137)?

Reviewer 2 Report

The authors managed to improve the manuscript, including in detail the cel viability assay. Hence, I suggest the acceptance of the manuscript. 

Author Response

Thanks for the reviewer’s carefully review.

Reviewer 3 Report

In this study, authors have isolated an amino acid derivative and an isovaleric acid phenethyl ester from a medicinal fungus. They have univocally characterised them by IR, HRESIMS, NMR (1H and COSY/HMBC, 13C and DEPT/HSQC) and ECD. These two compounds presented valuable neuroprotective properties against oxygen and glucose deprivation, and reduction of apoptosis in PC12 cells.

These results are of great relevance for discovery of neuroprotective drugs. I recommend it for publication in this journal. However, some details are missing and some changes should be done:

1.      Since cytotoxicity is an important parameter in drug development, did authors study cytotoxicity of compounds 1 and 2 over control PC12 cells? If so, please include the results.

2.      Regarding Figure 6, could authors provide numeric results (mean ± SD) belonging to cell viability?

3.      Since compounds 1 and 2 are dissolved in DMSO, did authors statistically compare DMSO and different compound (1 and 2) concentrations effect over PC12 cell viability against OGD? If so, please include the results.

4.      Authors stated that only compound 2 reduce the percentage of apoptotic cells at 1 µmol L-1, but in Figure 6 compound 1 presented even better results regarding cell viability. Could authors explain this different behaviour between compound 1 and compound 2?

5.      Please, according to IUPAC recommendation, change “µmol/L” for “µmol L-1”, “cells/well” for “cells well-1”, and so on, throughout the manuscript.

6.      Please, fill Acknowledgements section properly.

7.      Could author provide more recent bibliography? For example, provide replacements for [1-3] references.

Round 2

Reviewer 1 Report

There are many unresolved issues. The bioassay part is poorly planned. The experimental design (including the choice of microscopy technique) does not support the claimed effect of the compounds.

1. Please clarify all terms in Fig 6 - control, model, DMSO (how many %)?

2. The number of cells in different treatments does not seem comparable. The resolution is simply too low. It is difficult to make any conclusions from poor image quality.

3.Lines 136-137: " compound 2 (1 μmol L-1) can obviously reduce the percentage of apoptotic cells..." this statement is unsupported by any quantitative data. How did the authors conclude that? Conclusions must be supported by clear data. Quantitative data are needed. Similar comments for lines 239-240.

4. The claimed neuroprotective activities of the compounds are not supported by sufficient data. 

5, The authors also mentioned Tunnel staining but no data was given.

6. Instead of the ion current chromatograms, please provide PDA and TIC of EA extracts of both blank rice media and rice media + fungal biomass.

Reviewer 3 Report

I recommend it for publication.

Author Response

(The authors gave the same response as above.)
